# An Augmented Reality (AR) App Enhances the Pulmonary Function and Potency/Feasibility of Perioperative Rehabilitation in Patients Undergoing Orthopedic Surgery

**DOI:** 10.3390/ijerph20010648

**Published:** 2022-12-30

**Authors:** Pin-Hsuan Wang, Yi-Jen Wang, Yu-Wei Chen, Po-Ting Hsu, Ying-Ying Yang

**Affiliations:** 1Department of Medical Education, Clinical Innovation Center, Taipei Veterans General Hospital, Taipei 112, Taiwan; 2College of Medicine, National Yang Ming Chiao Tung University, Taipei 112, Taiwan; 3Department of Family Medicine, Taipei Veterans General Hospital, Taipei 112, Taiwan

**Keywords:** augmented reality, perioperative rehabilitation, pulmonary function, feasibility, potency, orthopedic surgery

## Abstract

Perioperative rehabilitation is crucial for patients receiving surgery in order to reduce complications and mortality. Conventional methods such as verbal instructions and pre-recorded video are commonly used, but several disadvantages exist. Therefore, we developed an augmented reality (AR) app that includes respiration training, resistance muscle training, and walking training for surgery preparation. The aim of this pilot study was to compare the effects of AR-based training rehabilitation programs with conventional (non-AR-based) programs considering the objective pulmonary function and subjective feasibility and potency in orthopedic patients. This prospective study was conducted in a medical center in Taiwan between 2018 to 2021. Sixty-six patients undergoing elective orthopedic surgery were allocated with a 1:1 ratio to non-AR or AR groups according to their wishes. After training, the inspiratory flow rate of the AR group was higher than that of the non-AR group pre-operatively. As for the subjective assessment, the feasibility (level of confidence and anxiety reduction) and potency (cooperation and educative effect) were superior in AR-based training, compared with the conventional training model. Our study showed that patients using our AR app had better subjective and objective outcomes compared with a conventional model for perioperative rehabilitation.

## 1. Introduction

The population of Taiwan is aging. Taiwan reached an aging society in 1993, aged society in 2018, and will reach super-aged society in 2025. In 2022, people aged 85 and over accounted for 10.5% of the total population [1]. As a population rapidly ages, the demand for orthopedic surgery also rises, especially joint replacement (e.g., total knee arthroplasty and total hip arthroplasty) [2]. A nationally representative study reported the incidence of total knee replacement consistently increased from 1996, especially among the elderly, with the highest demand being among those aged 70–79 years [3].

Studies have shown that the elderly population undergoing orthopedic surgeries experiences higher percentages of complications and morbidity compared with their younger counterparts [4,5,6]. In addition to common complications including bleeding, thromboembolic events, nerve injury, and implant loosening [7], pulmonary complications may occur [8]. Postoperative pulmonary complication (PPC) is defined as respiratory complications occurring within 48–72 hours after surgery [9], such as atelectasis, pneumonia, bronchitis, and bronchospasm [10]. One prospective cohort study from China showed the total incidence of early PPCs (post-operative 5–7 days) after total knee arthropathy was 45.9% [11]. Preoperative risks of PPC include older age (>65 years), smoking, chronic obstructive pulmonary disease (COPD), and bronchial asthma [9]. PPCs may increase morbidity and mortality, lengthen hospital stay, and further increase the health insurance burden [12].

Perioperative rehabilitation is crucial for patients receiving surgery to reduce complications and mortality [13]. In light of reducing PPCs, a guideline from the American College of Physicians suggests deep breathing exercises or incentive spirometry as postoperative interventions to be administered [14]. Preoperative rehabilitation including aerobic exercise, breathing exercise, and inspiratory muscle training has also been recommended [15]. The rehabilitation mentioned above could be delivered via health education leaflets, pre-recorded videos, verbal instruction, one-to-one therapist supervision, or onsite training [16]. However, there are some disadvantages of these conventional programs, including a low retention of the educational content, lack of supervision or accuracy, inconvenience for rural residents, and lack of motivation [16].

Because of the pace of technological innovation, virtual reality (VR) and augmented reality (AR) have also been introduced to the medical and physical fields [17]. VR immerses the patient into a simulated virtual world. When it is applied to rehabilitation, it could provide entertaining environments for patients to regain physical and fitness functions [18,19].

As for augmented reality (AR), it is a new technology involving seamless integration with virtual images in the real world. It could further align virtual objects with real world items, thus conducting a real-time experience [20]. AR is gradually applied to several fields, such as education, healthcare, construction, and entertainment, helping to enhance presence, liveness, and create a better experience [21].

The advantages of AR in the rehabilitation field are as follows. It could introduce gamification to rehabilitations, defined as “in non-game category using gaming design”, to arouse interest and motivation in patients [22]. It has also been reported that AR could increase engagement, increase overall interactivity, and prevent monotony and boredom by creating immersive patient experiences [23]. Real-time visual and auditory feedback provided by the AR may speed up and smooth rehabilitation by timely correcting patients’ exercise or movements, which further improve the effectiveness of rehabilitation [24]. Moreover, because of the Coronavirus disease 2019 (COVID-19) pandemic, the need for telemedicine is increasing [25]. AR is not limited by time or space, so it saves time for healthcare providers. The reason behind using an augmented reality instead of a virtual reality is that AR superiorly merges virtual environments with the real settings, creating more interaction with real life, thus providing a safer environment [26].

Therefore, we developed an innovative augmented reality (AR) app that includes respiration training, resistance muscle training, and walking training for surgery preparation. The aims of this study were as follows: (1) exploring the impact of AR-based training programs as perioperative rehabilitation on objective pulmonary function, and (2) exploring the subjective feasibility and potency of AR-based training programs in comparison with conventional perioperative rehabilitation programs among patients undergoing orthopedic surgery.

## 2. Materials and Methods

### 2.1. Study Design and Setting

This was a prospective, two-arm (AR group and non-AR group), non-randomized pilot study with a 1:1 allocation ratio. The objective outcome measure was repeated at three time points: preoperative day, postoperative day 1, and the day before discharge; subjective outcomes were assessed before and after the intervention. The study was conducted in a 2800-bed, 6000-staff medical center in northern Taiwan from April 2018 to December 2021.

### 2.2. Intervention

For patients in the AR group, an AR app was offered. It included 10 kinds of respiration training, 34 kinds of resistance muscle training (including 6 upper limbs strengthening exercises and 28 lower limbs strengthening exercises), and walking training. These programs were designed for individuals scheduled for elective orthopedic surgery, and the patients could present with or without chronic heart failure, myocardial infarction, degenerative joint disease, or chronic obstructive pulmonary disease. Healthcare providers (i.e., physiatrist, physical therapist, and nurses) were trained to choose a suitable training program for the patients.

The training program was led by an AR virtual teacher on the screen of mobile smart devices (e.g., smartphone or tablet). Patients could check the accuracy of their exercise on the screen and monitor their heart rate by wearing a specific bracelet. Meanwhile, the AR virtual teacher would recommend either an intermittent or continuous mode of walking training, and remind patients to speed up or slow down according to the results and special positions of the patient derived from the app. The AR app assists rehabilitation in multiple modalities, including constructing interactive, entertaining, and automatic therapeutic tools for rehabilitation. In order to motivate patients to train regularly, the AR system will give incentive points according to their completeness of the pre-set program. As the incentive points accumulated, the AR virtual teacher would upgrade from the first level to the fifth level, thus inspiring the patients. Patients and physicians could receive instant feedback on the data analysis. The completeness and achievement of all the patients’ training programs would be recorded in their app. The patient information was all anonymous. All of the patients would log into the app with a corresponding number without entering personal information. All of the recorded data could be retrieved anytime, for patient self-evaluation and patient–doctor interaction. During the follow-up appointment, the physician could sign in and check the patient’s data, including the completeness of the program. The app is free and available on both Google Play (Android) and Appstore (Apple) and can be downloaded by anyone who needs it.

As for the non-AR group, a pre-recorded video was provided to patients who chose conventional perioperative rehabilitation. The content included respiration training, resistance muscle training, and walking training. It was all pre-recorded, and the content was fixed, not personalized.

Both interventions were offered preoperatively. Rehabilitation utilizing the designated tool before and after the surgery was required.

### 2.3. Participants

Patients who met the following inclusion criteria were enrolled: (1) scheduled for elective orthopedic surgery, (2) needed enhancement of cardiopulmonary function, (3) aged 18 to 80 years, and (4) able to undergo spirometry measurement.

The exclusion criteria were as follows: (1) visual impairment or hearing impairment; (2) poor self-care or in bed-ridden status; (3) had no mobile smart device, e.g., smartphone, tablet, or computer; and (4) already had other ongoing rehabilitation programs. The research flowchart is shown in Figure 1.

### 2.4. Ethical Statement

Ethical approval (IRB no. 2018-07-030AC) was granted by the ethics committee of Taipei Veteran General Hospital, Taiwan. All of the enrolled patients were informed about the importance and advantage of this intervention for their safety. Oral consent was obtained from all patients. Questionnaire data were collected anonymously.

### 2.5. Outcome Assessment

Demographic data on age, gender, and years of education of the patients were collected at baseline. Objective and subjective outcomes of the perioperative rehabilitation were respectively assessed. The inspiratory flow rate was measured to objectively determine the pulmonary function using flow-oriented incentive spirometer (TriFlow device). This device had three chambers with one ball in each chamber. As air entered each chamber, the balls raised depending on the air flow inhaled per second. The corresponding inspiratory flow rate of each chamber was 600 milliliters per second (mL/s), 900 ml/s, and 1200 mL/s, respectively. The inspiratory flow rate was assessed at three time points: preoperative day, postoperative day 1, and the day before discharge from hospital, following one designated training program at each time point. The exact number of each patient’s inspiratory flow rate was recorded by the healthcare providers on a Google Sheet. The researchers then downloaded the whole google sheet for statistical analysis.

As for the subjective assessment, we utilized a self-administered questionnaire (Table 1) to obtain the patients’ views on the training program in two main domains—feasibility (familiarity, confidence, and anxiety) and potency (cooperation, educative effect, and accuracy). The level of agreement of each item was specified using a five-point agreement scale. The questionnaire was administered twice, once before and once after the rehabilitation program.

Furthermore, any reporting of adverse effects was recorded. A descriptive feedback of the experience was also collected from the enrolled patients, as well as the involved healthcare providers.

### 2.6. Data Analysis

Descriptive statistics and frequencies were used to characterize the study participants. A Chi-square test was used to compare the subjective assessment using a questionnaire, between AR and non-AR groups. The Chi-square test was used to compare the inspiratory flow rate at each of the three time points between the AR and non-AR groups. Statistical analyses were performed using IBM SPSS Statistics for Macintosh, Version 24.0 (IBM Corp., Armonk, NY, USA). A *p* value < 0.05 was considered statistically significant. Descriptive feedback was analyzed by identifying the themes from the contexts.

## 3. Results

### 3.1. Demographic Characteristics

Between 2018 and 2021, a total of 66 patients scheduled for elective orthopedic surgery participated in the study (*n* = 33 AR group, *n* = 33 non-AR group). There was a higher percentage of female enrolled in both the AR and non-AR groups (65% and 55%, respectively) (Figure 2a). The patients enrolled were mostly the elderly (≥65 years: 63% in both groups), and of a low educational level (≤9 years of education: 55% in both groups). Yet, the distribution of age and total years of education were similar between AR and non-AR group (Figure 2b,c).

### 3.2. Objective Outcome

The incentive spirometry ranked the patients’ inspiratory flow rate as 600, 900, or 1200 mL/s, from low to high. Owing to its nonparametric distribution, we used the Chi-square test for the statistical analysis. In Figure 3, the AR group performed better than the non-AR group for the first time point (preoperative day) (*p* = 0.009). The second time point (post-operative day one) showed no statistical differences between the two groups. As for the third (the day before discharge) time point, it showed a marginal statistical difference between them (*p* = 0.07).

### 3.3. Subjective Assessment

There was no statistical difference in the patients’ perceived familiarity with the rehabilitation program between AR-based and non-AR-based training, either before or after the training program (Figure 4a). As for levels of confidence, (less) anxiety, and cooperation, the AR-based training group was superior to the non-AR-based group after the training, but not statistically significant (Figure 4b,c).

The study also showed that the AR group had a more educative effect than the non-AR group after the training, but the difference was not statistically significant (Figure 5b). The information accuracy (Figure 5c) did not differ between the AR and non-AR groups, both pre- and post-training. No adverse effect was reported in both groups.

### 3.4. Descriptive Feedback

The following is feedback given by two of the patients and two of the healthcare providers. A 78-year-old woman, who had hypertension and type 2 diabetes mellitus, was enrolled in the program. After using the AR app, she mentioned “The app was very intriguing! I underwent an individual program and cooperated with my healthcare provider very well. My family and I strongly recommend this app to those with similar conditions”.

Another patient, a 75-year-old man with COPD, was encouraged by the AR app training, stating “After discussing with my healthcare provider, a customized exercise plan including muscle strengthening and walking training seemed very suitable. The AR app made me more energetic and enthusiastic! I believe that continuous practice using it could greatly improve my cardiopulmonary function”.

One of the healthcare providers mentioned the following, “After using the AR app, it seemed that the patients could understand how to appropriately inhale better, which made them more confident with further rehabilitation, and eventually lessened the staffs’ burden”. Another orthopedic ward staff commented “The AR app was convenient, user-friendly, and most importantly customized! Patients could select a suitable training program based on their own needs. The app could also correctly instruct the patient to perform proper range of motion of the joints. It was also worth mentioning that health providers could monitor the patients’ condition and progress to enhance compliance. If there was any error in the app, the patient could contact the health provider via the app immediately to figure out the solution”.

After analyzing the feedback from the patients and healthcare providers, three emerging themes around the advantages of the AR app were found, namely: (1) patients can repeat their training programs without the limitation of time and place, (2) the app could recognize whether the patient exercised in the right way, and (3) it could further facilitate regular exercise habits among patients.

## 4. Discussion

This prospective study found that pulmonary function inferred by the inspiratory flow rate was better in patients undergoing orthopedic surgery using the AR-based perioperative rehabilitation program compared with that of the conventional program, both on the preoperative day. The perceived feasibility (levels of confidence and anxiety) and potency (cooperation and educative effect) of AR-based training were superior to conventional training, although they were not statistically significant.

The analysis of this study was comprehensive. We documented objective data regarding the inspiratory flow rate, as well as a subjective assessment of the perceived feasibility (familiarity, confidence, and anxiety) and perceived potency (cooperation, educative effect, and accuracy). Descriptive feedback from both patients and healthcare providers was also recorded.

Perioperative rehabilitation is gaining importance nowadays. Many studies have shown AR to be effective in stroke rehabilitation [27], physical rehabilitation [28], gait impairment rehabilitation [29], cardiopulmonary rehabilitation [30], etc. Creating a user-friendly AR platform for perioperative exercise programs is a win–win. It not only benefits the patients, but also benefits the healthcare providers.

Although the subjective evaluation reported by the patients in the current study showed no statistical significance in both groups, the AR app could be regarded as a supplement therapy, aiding in ameliorating the rehabilitation process. Moreover, reports have shown that attentive functioning declines during aging [31], and the elderly are easily bored by traditional programs [17]. A gaming form of training could challenge and motivate patients, and also entertain them in the meanwhile [32]. “Technophobia”, defined as a “fear of technology”, is prevalent in the elderly [33]. While our study population was mostly the elderly and of a low educational level, the subjective result after AR training was shown to be similar to the conventional training model (pre-recorded video), indicating the accessibility of AR in elderly rehabilitation. Furthermore, with audiovisual aid, it could help reduce sports injury brought about by incorrect movement [34]. It could also create positive visual feedback, conduct a real-time experience, and bring patients a simulated sense of having a coach/teacher’s supervision.

Moreover, because of the COVID-19 pandemic, the need for telemedicine is growing [35]. Utilizing an AR app seems to be a crucial intervention for labor cost reduction, efficiency and quality in care, eco-friendliness, and public health enhancement [36]. Traditionally, perioperative training or rehabilitation programs are delivered via pre-recorded videos or written instructions. This AR app may be promising for boosting interactions and increasing motivation among patients during perioperative rehabilitation. The results of the current study can be strengthened by the positive experiences reported by the patients and healthcare providers.

### Limitation of the Study

There were several limitations in this study. The study sample in each group was relatively small; hence, further studies with a larger sample size are needed. Patients were self-allocated to each intervention group, which may have generated allocation bias, although there was no significant difference in characteristic distribution noted in this study. The subjective data were collected via questionnaires, which possibly presented questionnaire bias, especially when asking personalized and hypothetical questions. Moreover, blinding patients on the type of intervention they received was methodologically impossible. The assessors were also not blinded.

Moreover, one study showed that it took months to years to improve through rehabilitation after the main surgery [37], but we only recorded the data during hospitalization (1–2 weeks). In the case of longer rehabilitation with more practice, the patient may present better pulmonary function with better experience towards the AR app. We also hope that the AR app could be introduced to the patients earlier, that is as prehabilitation during outpatient visits as an extended perioperative rehabilitation.

Another limitation was that this study focused on pulmonary function indicated by the inspiratory flow rate. Indeed, the AR app had more options for patients, including upper and lower limb exercise, and bodyweight training. We look forward to focussing on strength training in the future.

## 5. Conclusions

This prospective study suggests that our AR app may add benefit to perioperative rehabilitation in orthopedic patients by enhancing pulmonary function. The feasibility (confidence and anxiety reduction) and potency (cooperation and educative effect) of the app are also promising. Our findings indicate the potential that AR may be a potential tool for engaging patients undergoing orthopedic surgery in physical activity, even in older adults.

## Figures and Tables

**Figure 1 ijerph-20-00648-f001:**
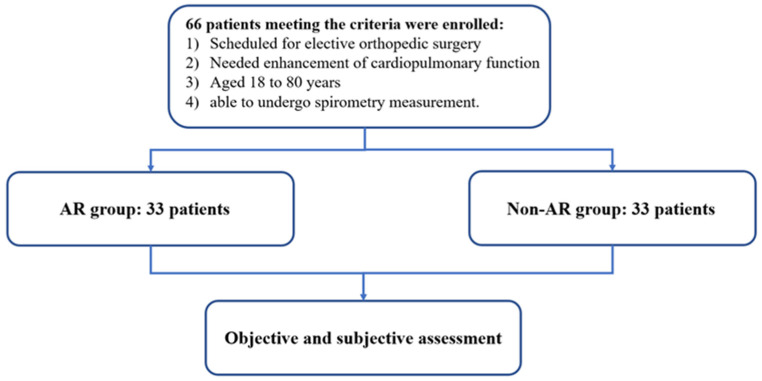
Flowchart of the recruitment of patients. Here, 66 patients were enrolled who met the following criteria: (1) scheduled for elective orthopedic surgery, (2) needed enhancement of cardiopulmonary function, (3) aged 18 to 80 years, and (4) able to undergo spirometry measurement. They were allocated to one of two groups (AR group and non-AR group) with 1:1 non-randomization.

**Figure 2 ijerph-20-00648-f002:**
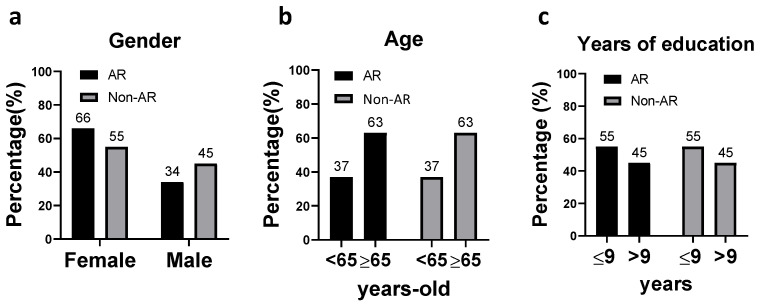
Distribution of the demographic characteristics in the study population: (**a**) gender, (**b**) age, and (**c**) years of education.

**Figure 3 ijerph-20-00648-f003:**
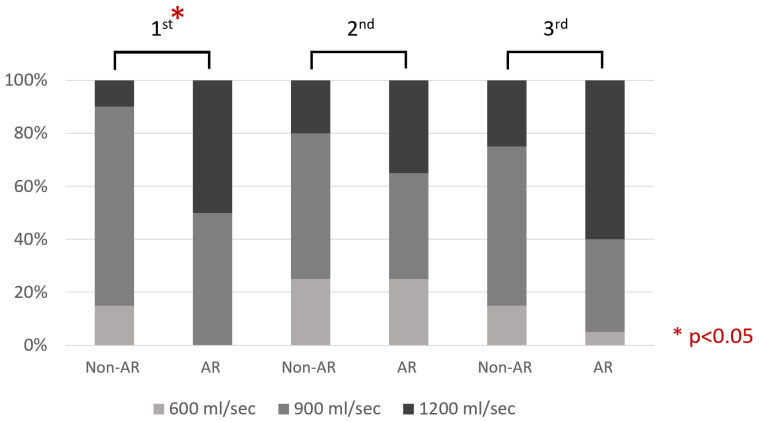
The distribution of both group’s inspiratory flow rate at different time points.

**Figure 4 ijerph-20-00648-f004:**
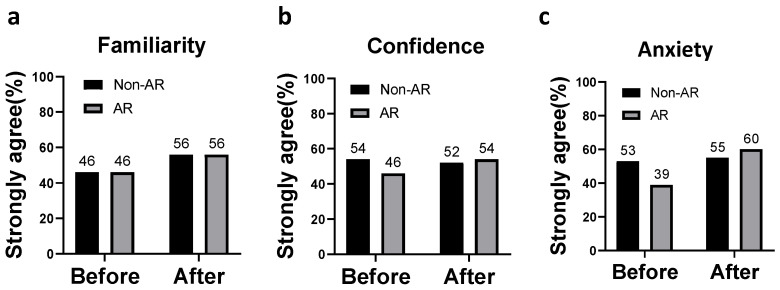
Perceived feasibility of the perioperative rehabilitation program before and after the designated program: (**a**) familiarity; (**b**) confidence; (**c**) anxiety.

**Figure 5 ijerph-20-00648-f005:**
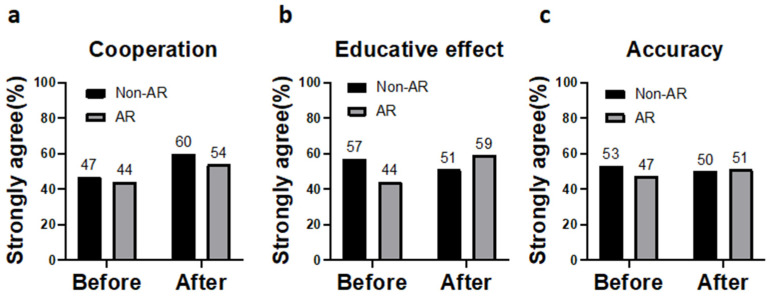
Perceived potency of the perioperative rehabilitation program before and after the designated program: (**a**) cooperation, (**b**) educative effect, and (**c**) accuracy.

**Table 1 ijerph-20-00648-t001:** Questionnaire items used to assess the perceived feasibility and perceived potency in this study.

Domain	Item	Statement of Item	Agreement Scale
Feasibility	Familiarity	Are you familiar with the rehabilitation program?	5: strongly agree 4: agree 3: neutral 2: disagree 1: strongly disagree
	Confidence	Are you confident with the rehabilitation program?	(same as above)
	Anxiety	Are you anxious about not being able to finish the rehabilitation program?	(same as above)
Potency	Cooperation	Are you willing to cooperate with the healthcare providers to finish the rehabilitation program?	(same as above)
	Educative effect	Do you agree that the rehabilitation program is educative?	(same as above)
	Accuracy	Do you agree that the rehabilitation program can precisely convey information?	(same as above)

## Data Availability

The data presented in this study are available upon request from the corresponding author. The data are not publicly available due to privacy and ethical considerations.

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
