# Peer review of "An Augmented Reality (AR) App Enhances the Pulmonary Function and Potency/Feasibility of Perioperative Rehabilitation in Patients Undergoing Orthopedic Surgery"

_ijerph, 2022, doi:10.3390/ijerph20010648_

Round 1

Reviewer 1 Report

The proposed manuscript presents a study where the effect of using augmented reality technologies in a perioperative rehabilitation context has been evaluated. Specifically, the authors focus on diminishing pulmonary complications as this is an issue that can happen after an orthopedic surgery, especially in the elderly. The proposal focuses on improving the pulmonary function in a perioperative rehabilitation context. As concluding remarks, it is mentioned that the results of this study suggests that the augmented reality app is beneficial to enhance pulmonary functions. However, this reviewer has some concerns about the methodology and the result. Firstly, I suggest to detail the properties and functionalities of the application as it is assumed that not all the augmented reality applications are able to reach this result. Indeed, the design and final app surely are responsible in the final contribution of the study. Secondly, about the results, not only the graphs but also the data is useful to understand what is happening. A more detail comments about these data and the corresponding descriptive statistics could be also of use for the reader. Additionally, these data should be included considering the different groups of sex, age, … previously defined.

Other comments:

The percentage of people suffering from pulmonary complications after undergoing orthopedic surgeries should be included.

The last but one paragraph of the introduction lists some advantages of the augmented reality technology, but they can also be applicable to virtual reality. The reasons behind using augmented reality instead of virtual reality could be indicated.  

The path through the activities suggested by the specialist in the augmented reality application are patient specific. However, in the counterpart, the material and videos for the traditional way are generic and valid for all the patients.

Do the authors have any data about the usage of the material for the participants in the non-AR group?

About figure 2c, what are the reasons behind the decline on the flow for the AR group from step 1 to 2?

Minor comment:

The first sentence in the section “Participants” sounds a bit strange to this reviewer.

Author Response

[IJERPH] Manuscript ID: ijerph-2036022 "Augmented reality (AR) app enhances pulmonary function and potency/feasibility of perioperative rehabilitation in patients undergoing orthopedic surgery"

Point-to-point responses to Reviewer 1:
Comment 1: I suggest to detail the properties and functionalities of the application as it is assumed that not all the augmented reality applications are able to reach this result. Indeed, the design and final app surely are responsible in the final contribution of the study.

Response 1: Thank you for the suggestion and comments! We really appreciate that. We had detailed the properties and functionalities of the augmented reality application in 2.2 Intervention (page 3, line 100- 126).

Comment 2: About the results, not only the graphs but also the data is useful to understand what is happening. A more detail comments about these data and the corresponding descriptive statistics could be also of use for the reader.

Response 2: Thank you for the suggestion. We added more detail comments about these data and the corresponding descriptive statistics. We revised the manuscript, and listed in page 5, line 195- 197.

Comment 3: These data should be included considering the different groups of sex, age, … previously defined.

Response 3: Thank you for the suggestion! However, we did not have such data during the study.

Comment 4: The percentage of people suffering from pulmonary complications after undergoing orthopedic surgeries should be included.

Response 4: Thank you for the suggestion! There is no accurate data about PPC after undergoing orthopedic surgeries. However, one prospective cohort study from China showed the total incidence of early PPCs (post-operative 5-7 days) after total knee arthropathy was 45.9%. We revised the manuscript, and we had listed in page 2, line 46- 48, and the following reference was also added.

Comment 5: The last but one paragraph of the introduction lists some advantages of the augmented reality technology, but they can also be applicable to virtual reality. The reasons behind using augmented reality instead of virtual reality could be indicated. 
Response 5
: Thank you for the suggestion! The reason behind using augmented reality instead of virtual reality is that AR superiorly merges virtual environments with the real settings, creating more interaction with real life, providing a safer environment. We revised the manuscript, and we had listed in page 2, line 81-83, and the relevant reference was also added.

Comment 6: The path through the activities suggested by the specialist in the augmented reality application are patient specific. However, in the counterpart, the material and videos for the traditional way are generic and valid for all the patients.
Response 6:
Thanks for the comments. In fact, the augmented reality application we created was not patient specific. We hope that it could be both generic and valid for all the patients, just like the conventional materials and videos.  

Comment 7: Do the authors have any data about the usage of the material for the participants in the non-AR group?
Response 7:
Thanks for the suggestions. We do not provide such materials during the study.

Comment 8: About figure 2c, what are the reasons behind the decline on the flow for the AR group from step 1 to 2?
Response 8:
Each time point of Figure 2c was pre-operative day one, post-operative day one and the day before discharge. The reason why the flow showed decreased for in the AR group might be secondary to pain, post-operative fatigue, delay recovery from analgesia. We had revised the manuscript and listed in page 5, line 195- 197.

Comment 9: The first sentence in the section “Participants” sounds a bit strange to this reviewer.
Response 9:
Thank you for pointing out the grammatical error, we had revised the first sentence as “Patients met the following inclusion criteria 

Reviewer 2 Report

This work presents a useful information to be consider when a AR project will be design for a rehabilitation propose, however my sugesstion for future related project is to present more references with AR .

The senarios for test, could be design according the age patient.

Specific Comments:

1. A main question is not described directly, although is represented at the introduction section, where a perioperative rehabilitation is crucial to patients receiving surgery to reduce complica-13 tions and mortality.

2. The topic of this paper is relevant ,because of the increased use of telemedicine due the COVID19, the AR app seemed to be one crucial intervention to reach a wide range of patients, at the same time AR permit a cost reduction on health process.

3. Compared with other published material, this work used the AR to permit the patient learn about rehabilitation process and at the same time improve their own rehabilitation.

4. In terms of methodology, could include a flow that reflect the methodology steps.

5. The main question is answered indirectly using the conclusions section to resume the fact.

6. The cited references are appropiate, realated with health and VR topics.

  •  
  •  

Author Response

[IJERPH] Manuscript ID: ijerph-2036022 "Augmented reality (AR) app enhances pulmonary function and potency/feasibility of perioperative rehabilitation in patients undergoing orthopedic surgery"

Point-to-point responses to Reviewer 2:

Comment 1: A main question is not described directly, although is represented at the introduction section, where a perioperative rehabilitation is crucial to patients receiving surgery to reduce complications and mortality.
Response 1:
Thanks for the suggestion. We revised the manuscript, and listed in page 2, line 51- 52, and we also revised the relevant reference.

Comment 2: The topic of this paper is relevant, because of the increased use of telemedicine due the COVID19, the AR app seemed to be one crucial intervention to reach a wide range of patients, at the same time AR permit a cost reduction on health process.

Response 2: Thanks for the comments. We really appreciate that.

Comment 3: Compared with other published material, this work used the AR to permit the patient learn about rehabilitation process and at the same time improve their own rehabilitation.

Response 3: Thanks for the comments. We really appreciate that.

Comment 4: In terms of methodology, could include a flow that reflect the methodology steps.

Response 4: Thanks for the suggestion. We had provided a flowchart of the recruitment of patients, and we labeled as Figure 1. Figure itself and the legend were both attached to the file. We also revised the manuscript, and was listed in page 3, line 139. We also revised other figure labels subsequently.

Comment 5: The main question is answered indirectly using the conclusions section to resume the fact.

Response 5: Thanks for the suggestion. Our study did not collect such data, so the main question could not be answered directly.

Comment 6: The cited references are appropriate, related with health and VR topics.

Response 6: Thanks for the comments. We really appreciate that.

Round 2

Reviewer 1 Report

Thank to the authors for the improvement in the version. However, This reviewer already has some concerns: 

- Data. For instance, figure 3 is including the mean, but a statistical description of the original data including, minimum, maximum, median, quartiles is not provided. 

- Another important concern is about the conclusions. The authors indicate "This prospective study found that pulmonary function inferred by the inspiratory 255 flow rate was better in patients undergoing orthopedic surgery using the AR-based peri-256 operative rehabilitation program compared to that of the conventional program...", however, looking to the graph in Figure 3, it illustrates that the day before being discharged is better in comparison to the entry date in relation with the AR counterpart. It would be interesting observing these differences with data and by performing a stronger analysis. 

Author Response

[IJERPH] Manuscript ID: ijerph-2036022 "Augmented reality (AR) app enhances pulmonary function and potency/feasibility of perioperative rehabilitation in patients undergoing orthopedic surgery"

Dear Prof. Dr. Paul B. Tchounwou

Thank you for your e-mail dated December 22th, 2022. Thank you for your comments and suggestion. Please help us remove three co-authors [professor Chen-Huan Chen , Yung-Yang Lin , Chi-Hung Lin] from the submission websites. The authorship changed form had been signed by all authors and attacked as below.

Point-to-point responses:

Comment 1: English language and style are fine/minor spell check required

Response 1: Thank you for the suggestion. We had carefully checked the English language and style. We corrected the grammatical error and revised it accordingly.

Comment 2: Data. For instance, figure 3 is including the mean, but a statistical description of the original data including, minimum, maximum, median, quartiles is not provided.

Response 2: Thank you for the suggestion and comments! After consulting our statistician, considering objective assessment, we corrected the statistical methods from student’s-t test to chi-square test because the outcome measure was categorical, but not continuous variable. We provided a new Figure 3, illustrating the distribution of both group’s inspiratory flow rate at different time points, which is slightly different from our prior version. AR group performed better than the non-AR group for the first time point (preoperative day) (p= 0.009). We revised the manuscript accordingly on page 6, line 200-206.

Comment 2: Another important concern is about the conclusions. The authors indicate "This prospective study found that pulmonary function inferred by the inspirator flow rate was better in patients undergoing orthopedic surgery using the AR-based peri- operative rehabilitation program compared to that of the conventional program...", however, looking to the graph in Figure 3, it illustrates that the day before being discharged is better in comparison to the entry date in relation with the AR counterpart. It would be interesting observing these differences with data and by performing a stronger analysis.

Response 2: Thank you for the suggestion. Because our results are revised, the conclusion was also modified. We downplayed our conclusion according to the suggestion. We revised the manuscript, and listed in page 9, line 318- 322.
